

# An in vitro ovarian explant culture system to examine sex change in a hermaphroditic fish

Alexander Goikoetxea[1], Erin L. Damsteegt[2], Erica V. Todd[3], Andrew McNaughton[1], Neil J. Gemmell[1] and P. Mark Lokman[2]

[1] Department of Anatomy, University of Otago, Dunedin, New Zealand
[2] Department of Zoology, University of Otago, Dunedin, New Zealand
[3] School of Life and Environmental Science, Deakin University, Geelong, Australia

## ABSTRACT

Many teleost fishes undergo natural sex change, and elucidating the physiological and molecular controls of this process offers unique opportunities not only to develop methods of controlling sex in aquaculture settings, but to better understand vertebrate sexual development more broadly. Induction of sex change in some sequentially hermaphroditic or gonochoristic fish can be achieved in vivo through social manipulation, inhibition of aromatase activity, or steroid treatment. However, the induction of sex change in vitro has been largely unexplored. In this study, we established an in vitro culture system for ovarian explants in serum-free medium for a model sequential hermaphrodite, the New Zealand spotty wrasse (*Notolabrus celidotus*). This culture technique enabled evaluating the effect of various treatments with $17\beta$-estradiol ($E_2$), 11-ketotestosterone (11KT) or cortisol (CORT) on spotty wrasse ovarian architecture for 21 days. A quantitative approach to measuring the degree of ovarian atresia within histological images was also developed, using pixel-based machine learning software. Ovarian atresia likely due to culture was observed across all treatments including no-hormone controls, but was minimised with treatment of at least 10 ng/mL $E_2$. Neither 11KT nor CORT administration induced proliferation of spermatogonia (i.e., sex change) in the cultured ovaries indicating culture beyond 21 days may be needed to induce sex change in vitro. The in vitro gonadal culture and analysis systems established here enable future studies investigating the paracrine role of sex steroids, glucocorticoids and a variety of other factors during gonadal sex change in fish.

## INTRODUCTION

In most vertebrates, sex is fixed after birth and remains the same throughout life (*Piferrer & Guiguen, 2008*; *Barske & Capel, 2008*; *Todd et al., 2016*; *Liu et al., 2017*). Some teleost fishes, however, develop as one sex but remain able to change to the other during adulthood (*Godwin, 2009*; *Lamm et al., 2015*; *Liu et al., 2017*; *Gemmell et al., 2019*). In vitro culture manipulations enable precise monitoring of the effects of biotic and abiotic factors (e.g., temperature, pH, microplastics, etc.) on gonadal development, but are underutilised in aquaculture and fisheries research. Ex vivo approaches show particular potential to study

Corresponding author
Alexander Goikoetxea, alexander-goikoetxea@gmail.com

responses of gonadal tissue to sex steroids and other hormones (e.g., glucocorticoids). Testicular and ovarian organ culture has been experimentally achieved with a handful of gonochoristic (fixed separate sexes) fish species (*Carragher & Sumpter, 1990*; *Miura et al., 1991*; *Ozaki et al., 2006*; *Ozaki et al., 2019*; *Fernandino et al., 2012*). However, with the exception of a conference abstract on the effects of several hormones on the gonadal architecture of sex-changing three-spot wrasse (*Halichoeres trimaculatus* (*Todo et al., 2008*)), explant culture systems for the study of sex change in fish remain unreported. Gonadal remodelling in sex-changing fishes occurs in adulthood and, therefore, the study of sex-determining mechanisms is not limited to egg or larval stages as in other vertebrates. Tissue and organoid in vitro culture allows evaluation of multiple factors in a large number of replicates in well-controlled and standardised experimental conditions, while reducing animal usage and handling compared with in vivo manipulations (*Miura et al., 1991*; *Schulz et al., 2010*; *Ozaki et al., 2019*).

The aim of this study was to develop an organ culture system for a sequential hermaphrodite to establish a new model for investigating the molecular and cellular basis of sex reversal in the gonad. We chose the New Zealand spotty wrasse (*Notolabrus celidotus*), a locally common and widespread marine species, to evaluate the effects of a suite of hormonal factors on gonadal architecture. In this diandric protogynous hermaphroditic species, sex change is socially regulated, whereby the removal of the dominant male from the social group induces sex change in a resident female (*Thomas et al., 2019*). Spotty wrasses exhibit sexual dimorphism with alternative male phenotypes. Initial phase (IP) individuals consist of females and primary males which develop directly from a juvenile stage with a bipotential gonad (*Choat, 1965*). Both females and IP males can sex or role change, respectively, to become terminal phase (TP) males (*Choat, 1965*).

While natural sex change involves modifications at the behavioural, physiological, molecular and anatomical levels (*Todd et al., 2016*), herein, the term 'sex change' is used to refer to the development of testicular tissue in female gonads. The effect of different doses of steroid hormones 17$\beta$-estradiol (E$_2$), 11-ketotestosterone (11KT) or cortisol (CORT) on spotty wrasse ovary explants was evaluated in vitro to elucidate their effect on gonadal sex change in this species. Controls (C, no steroid) were used to determine the viability of the culture system and the potential triggering of gonadal sex change in the absence of exogenous factors. Treatment with E$_2$ was used to evaluate its role in the successful maintenance of spotty wrasse ovaries in vitro. 11KT, the most potent androgen in fish (*Goikoetxea, Todd & Gemmell, 2017*), was employed to explore its potential to induce sex change (i.e., proliferation of spermatogonia) in cultured spotty wrasse ovaries. In addition, treatment with the stress hormone CORT was predicted to induce oocyte degeneration through the inhibition of aromatase (*Fernandino et al., 2013*), and it was hypothesised that this effect alone may trigger the female-to-male gonadal transformation. A strong effect of CORT, the main glucocorticoid in fish, has been linked to fish sex determination and sex change in several studies (*Hayashi et al., 2010*; *Yamaguchi et al., 2010*; *Kitano et al., 2012*; *Nozu & Nakamura, 2015*; *Miller et al., 2019*). In gonochoristic species such as the Japanese flounder (*Paralichthys olivaceus*) or medaka (*Oryzias latipes*), in vivo CORT treatment has been reported to mediate masculinisation of genetic females by promoting

androgen production and apoptosis in the gonad, inducing maleness (*Hayashi et al., 2010*; *Yamaguchi et al., 2010*; *Kitano et al., 2012*). Likewise, recent studies suggest a pivotal role of CORT in driving natural sex change in sequentially hermaphroditic teleosts (*Nozu & Nakamura, 2015*; *Chen et al., 2020*). It has been hypothesised that CORT, together with genetic (e.g., *amh* upregulation) and epigenetic (e.g., DNA-methylation) factors could lead to suppression of the female genetic network in favour of enhancing male pathway gene expression to trigger sex change (*Todd et al., 2016*; *Liu et al., 2017*; *Gemmell et al., 2019*). This is the first study to evaluate the effects of CORT in an in vitro gonadal culture system of a sex-changing fish.

## MATERIALS AND METHODS

### Sample collection

Five IP spotty wrasse individuals ranging from 160–230 mm total length were captured by hook and line at the Portobello Marine Laboratory, Department of Marine Science, University of Otago, New Zealand during May 2019. Live fish were immediately transported to cell culture facilities at the Department of Zoology, University of Otago, where fish were euthanised by overdose in benzocaine (0.3 g/L) and gonads dissected within 1.5 h of capture. At this time, gonadal fragments from each fish were preserved for histological analysis as day 0 reference tissues (D0). Gonads fixed in 4% PFA were processed for routine embedding in paraffin (Otago Histology Services Unit, Department of Pathology, University of Otago). Fish were captured and manipulated with approval from the University of Otago Animal Ethics Committee (AUP-18-247), and in accordance with New Zealand National Animal Ethics Advisory Committee guidelines.

### Ovarian organ culture technique

In vitro culture of spotty wrasse ovary was carried out using the floating tissue culture method originally described by *Miura et al. (1991)* with minor modifications. For each of 5 IP fish, freshly removed ovaries were cut into pieces of approximately 2–3 mm and ovarian fragments were placed on floats of 1.5% agarose covered with a nitrocellulose membrane, each fragment in a well that further contained 500 µl of medium in a 24-well plastic tissue culture dish. The basal medium consisted of Leibovitz L-15 medium supplemented with 0.5% bovine serum albumin, 10,000 U/L penicillin, 10 mg/L streptomycin and 1 µg/mL porcine insulin adjusted to pH 7.4. Ovarian fragments were exposed to $E_2$, 11KT or CORT for 21 days at 16 °C. The effect of various concentrations of each hormone (1, 10 or 100 ng/mL) was examined. Steroids were first dissolved in EtOH and then diluted with the medium (0.1% EtOH), which was changed every 7 days. Controls (no steroid, 0.1% EtOH) were included. At the end of the experiment, tissues were fixed in 4% PFA and processed for routine paraffin embedding (Fig. S1).

### Image analysis

An open source pixel-based machine learning trainable classifier, Ilastik (v. 1.3.2) (https://www.ilastik.org/) (*Sommer et al., 2011*), was employed to discriminate cell types in the gonadal images captured using brightfield light microscopy. Approximately 85

previtellogenic oocytes (PVO), from a subset of 5 images, were used to train Ilastik. Atretic oocytes, stromal tissue and background were labelled separately with a paintbrush-style tool using the Pixel Classification module. The following feature selection was applied: colour/intensity, $\sigma_3 = 1.60$; edge, $\sigma_3 = 1.60$; texture, $\sigma_3 = 1.60$. The training process was refined iteratively, updating pixel classifications until object types were accurately discriminated in the segmentation maps, as described in *Logan et al. (2016)* (Fig. 1). The Ilastik classifier files were extracted in HDF5 (Hierarchical Data Format 5) format and loaded into open source image processing software ImageJ (v. 2.0.0-rc-69/1.52p), where the different object types were each assigned a different threshold. ImageJ 'Analyse particles' function was used to calculate the total relative surface area occupied by non-atretic PVO and to measure the total surface of the tissue section in each image. 'Remove outliers' function was used to remove any outlier <10 pixels from analysis and 'Fill holes' function was used to mark PVO nuclei in the exceptional occasions when Ilastik failed to correctly recognise these.

## Statistical analysis

Generalised linear mixed models were used to assess the effect of steroid (i.e., $E_2$, 11KT or CORT) treatments on the proportion of total tissue surface occupied by non-atretic PVO ('PVO_area') in comparison to the total area of each ovarian section ('total_area'), using individual fish as a random effect. Models followed a negative binomial distribution and were performed using the following command line in R (v. 3.6.1): 'glmmPQL (PVO_area/total_area ~Treatment, random = ~1|Fish, weights = total_area, family = negative.binomial (theta = 1), data = dat)'. If significant differences between treatments were found, Tukey's multiple comparison tests were performed to determine where the significance lay between treatments. All statistical analyses were performed using R (*R Core Team, 2013*). Results are expressed as means ± SEM of 5 biological replicates, except for the control ($n = 15$) or unless stated otherwise.

## RESULTS AND DISCUSSION

Histological analysis of D0 reference tissues confirmed that all 5 IP spotty wrasse captured were female. Visual inspection of gonadal sections under light microscopy revealed that oocyte degeneration was present in all experimental groups, including controls, but not in the D0 reference tissues. However, initiation of spermatogenesis was not observed in any group, and no male structures were identified.

Atresia of PVO appeared to be a background effect of the in vitro culture (*Todo et al., 2008*). This was reflected in reduced oocyte surface area in controls compared to D0 reference tissues, with the proportion of total tissue area filled with non-atretic PVO being significantly higher in D0 ($73.6 \pm 2.4\%$) versus control ovaries ($31.8 \pm 3.7\%$) ($p$-value < 0.001) (Fig. 2A). Estrogen treatment reduced the incidence of oocyte degeneration. A greater surface area covered by non-atretic PVO was observed in ovaries treated with the two highest $E_2$ doses (10 ng/mL, $51.1 \pm 5.1\%$; 100 ng/mL, $50.7 \pm 9.8\%$) (Fig. 2A). Differences between the control ovaries and those treated with 100 ng/mL of $E_2$ were significant ($p$-value < 0.05). Therefore, exogenous $E_2$ administration to cultured ovaries

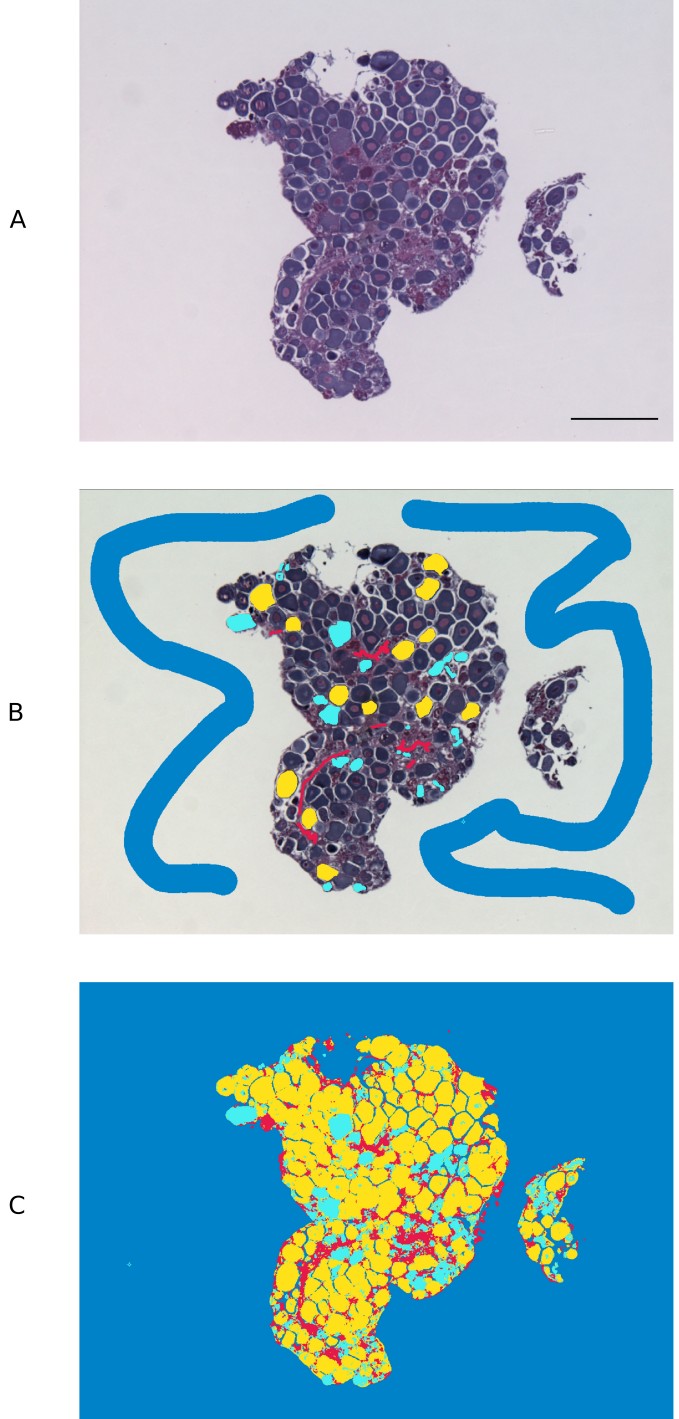

**Figure 1** **Pixel-based trainable classifier Ilastik workflow.** (A) Example of spotty wrasse (*Notolabrus celidotus*) ovarian tissue treated with 100 ng/mL of 17$\beta$-estradiol processed for histology and visualised under light microscopy. (B) Ilastik was manually trained to discriminate previtellogenic oocytes (yellow), atretic oocytes (turquoise), stromal/connective tissue (red) and background (blue). (C) Segmentation map with different object types created with Ilastik. Scale bar: 100 $\mu$m.

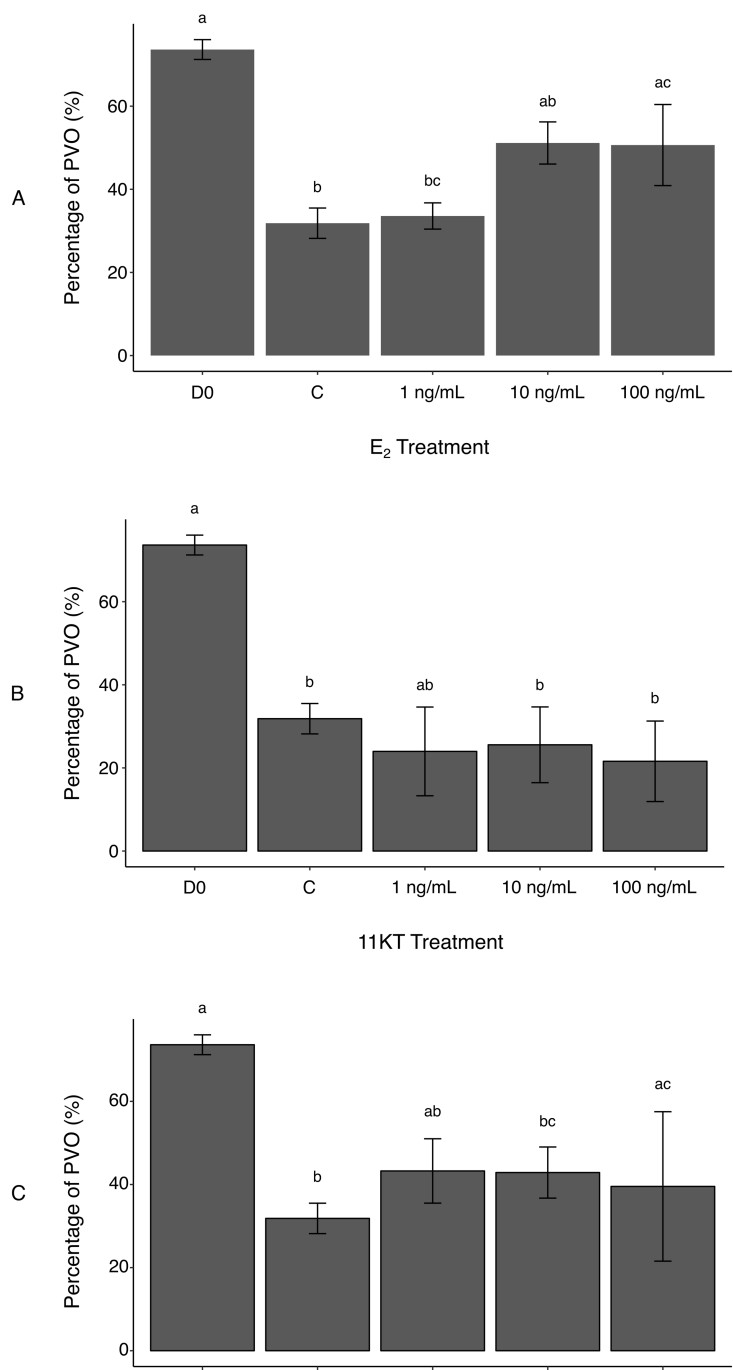

**Figure 2** **Histological measurement of area covered by non-atretic previtellogenic oocytes (PVO) in ovaries of spotty wrasse (*Notolabrus celidotus*) treated with E$_2$ (A), 11KT (B) or CORT (C) (day 21).** Bars show the surface area of all non-atretic PVO over total tissue area (expressed as %). Same day 0 reference tissues and controls (day 21) were used for all treatments. (continued on next page...)

**Figure 2 (…continued)**

Results are shown as the mean ± SEM. Different letters denote a statistically significant difference between groups, while experimental groups with no significant differences share the same letter. Samples sizes: D0 $n = 5$; C (no steroid) $n = 15$; $E_2$, 1 ng/mL $n = 5$, 10 ng/mL $n = 5$, 100 ng/mL $n = 5$; 11KT, 1 ng/mL $n = 5$, 10 ng/mL $n = 5$, 100 ng/mL $n = 5$; CORT, 1 ng/mL $n = 5$, 10 ng/mL $n = 5$, 100 ng/mL $n = 4$. Abbreviations: 11-ketotestosterone (11KT), control (C), cortisol (CORT), day 0 reference tissues (D0), 17$\beta$-estradiol ($E_2$).

can effectively, albeit partially, prevent oocyte degeneration (i.e., increased surface area occupied by non-atretic PVO) associated with explant culture. This reinforces observations in female coho salmon (*Oncorhynchus kisutch*), in which in vivo fadrozole treatment (an aromatase inhibitor) induced reductions in $E_2$ and a higher incidence of associated atresia (*Afonso et al., 1999*). In future culture systems, the addition of growth factors or the use of an artificial extracellular matrix (*Kim et al., 2011*) could be investigated to further mitigate oocyte degeneration as a consequence of culture.

Unexpectedly, there was no histological evidence that treatment with 11KT induced sex change in spotty wrasse ovaries in vitro. The proportion of non-atretic PVO found in the ovarian tissues was not dose-dependent and did not significantly vary between any of the 11KT-treated ovaries. Although all 11KT-treated ovary explants showed a much lower percentage of PVO compared to total tissue area than D0 ovaries, values were comparable to the control (Fig. 2B). The proportion of non-atretic PVO in 11KT-treated ovarian tissues was also unaffected by dose (1 ng/mL, 24.0 ± 10.7%; 10 ng/mL, 25.6 ± 9.1%; 100 ng/mL, 21.6 ± 9.7%).

Successful induction of spermatogenesis via in vitro androgen treatment in cultured testes has been demonstrated in Japanese eel (*Anguilla japonica*), in which Sertoli cells were activated by 11KT to stimulate the spermatogenic cascade (*Miura et al., 1991*; *Ozaki et al., 2006*). The role of 11KT as a promoter of spermatogenesis in ovaries of sex-changing fish has also been demonstrated (*Bhandari et al., 2006*; *Todo et al., 2008*). In the protogynous honeycomb grouper (*Epinephelus merra*), in vivo implantation of 11KT into pre-spawning females caused masculinisation in all fish (*Bhandari et al., 2006*). Moreover, in vivo induction of sex change was also accomplished in the bidirectional sex changing *Pseudolabrus sieboldi*, both by administering sustained-release capsules containing 11KT to females and those with $E_2$ to male individuals (*Ohta et al., 2012*). Implants containing 11KT induced changes in body colour in the females too (*Ohta et al., 2012*). Likewise, in vitro 11KT treatment of ovarian explants from protogynous three-spot wrasse induced the formation of presumed spermatogenic crypts and sperm within 2–3 weeks (*Todo et al., 2008*). However, three-spot wrasses live in tropical waters, and sex change in captive populations has been observed in under 40 days (*Kuwamura et al., 2007*), around half the time required in spotty wrasse (60–70 days (*Thomas et al., 2019*)). In the current study, it may be that spotty wrasse ovaries were not cultured long enough for 11KT to promote male germ cell proliferation, or that 11KT alone is insufficient to induce gonadal transformation in this species. It has been hypothesised that endogenous $E_2$ levels need to decrease below a physiological threshold for oocyte maintenance to become jeopardised and androgen levels to effectively induce proliferation of bipotential gonial

stem cells in the ovary (*Bhandari et al., 2006*). Co-treatment of 11KT with an aromatase inhibitor (e.g., fadrozole) should be performed in the future to further characterise the triggering of in vitro gonadal restructuring, potentially establishing this technique as a model for sex change research.

It was hypothesised that CORT treatment would promote oocyte degeneration, following evidence that CORT may mediate sex change in fish by inhibiting aromatase transcription and promoting androgen production (*Fernandino et al., 2012*; *Goikoetxea, Todd & Gemmell, 2017*). However, no significant differences were observed in the proportion of non-atretic PVO between the controls and ovaries treated with 1 and 10 ng/mL of CORT (43.3 ± 7.7% and 42.9 ± 6.2%, respectively) (Fig. 2C). While a significant increase in the percentage area of PVO was detected between the controls and 100 ng/mL CORT treatment group (39.5 ± 18.0%) (*p*-value <0.005), this may reflect variability between fish and the effect of smaller sample size ($n = 4$, as one biological replicate was lost during histological processing due to technical issues) on the statistical analysis. Therefore, these differences were not deemed a direct effect of CORT administration to the ovaries. Cortisol has been observed to negatively impact ovarian development by suppressing $E_2$ secretion in cultured rainbow trout (*Oncorhynchus mykiss*) follicles (*Carragher & Sumpter, 1990*). However, in that same study, Carragher & Sumpter (*Carragher & Sumpter, 1990*) argued that there may be a time lag for CORT suppressive effects to become evident during in vitro ovarian culture. Longer-term culture of spotty wrasse ovaries may be needed to observe any effects of CORT on ovarian structure, a finding in keeping with our findings for 11KT.

CORT failed to initiate obvious sex change in cultured spotty wrasse ovaries, with no evidence of male tissues observed in the ovarian sections. It may be that CORT alone is ineffective in inducing spotty wrasse sex change, or that the experiment needs technical adjustments (e.g., longer duration) for the successful induction of sex change in spotty wrasse. Low doses of CORT (i.e., 0.01–10 ng/mL) were reported to promote testicular development in Japanese eel and pejerrey (*Odontesthes bonariensis*) by increasing the production of 11KT, which, in turn, induced spermatogonial proliferation (*Ozaki et al., 2006*; *Fernandino et al., 2012*). Interestingly, higher doses of CORT (i.e., 100 ng/mL) had the opposite effect and inhibited 11KT synthesis in these species (*Ozaki et al., 2006*; *Fernandino et al., 2012*). Although measurement of 11KT concentration in the cultured media should be performed to evaluate such effects on spotty wrasse ovaries, absence of spermatogonia in the cultured ovaries suggests 11KT levels were not significantly affected. The possible metabolic conversion of CORT into 11KT and other 11-oxygenated androgens has also been suggested in several teleosts (*Kime, 1978*; *Ozaki et al., 2006*), as there is cross-talk between the glucocorticoid and androgen synthesis pathways (*Arterbery, Deitcher & Bass, 2010*; *Goikoetxea, Todd & Gemmell, 2017*). However, the mechanisms underlying 11KT synthesis are still not fully understood and a potential CORT-to-11KT direct conversion remains to be clarified (*Schulz, 1986*; *Goikoetxea, Todd & Gemmell, 2017*). This work constitutes the first-ever in vitro organ culture system in a temperate sex-changing teleost, and opens doors for future work in a wide variety of hermaphroditic fish.

Although the current experiment could not induce any observable female-to-male restructuring of spotty wrasse gonads in culture, ovarian tissues were successfully

maintained in culture for 21 days. Such long-term culture of fish organ explants has only been accomplished in a few instances (*Carragher & Sumpter, 1990*; *Miura et al., 1991*; *Ozaki et al., 2006*; *Ozaki et al., 2019*; *Fernandino et al., 2012*). Furthermore, this work validates the use of open-source machine learning software Ilastik to reliably recognise cell types in fish gonads, with enormous potential in future studies. The image analysis strategy presented here can be adapted and applied to other studies where accurate cell segmentation is required.

## CONCLUSION

This study developed an organ culture system for gonadal tissue explants in New Zealand spotty wrasse, providing an in vitro system for investigating control and perturbation of gonadal sex change in this and other species. Moreover, we successfully integrated Ilastik software to score ovarian architecture, an application which provides great scope for future studies on effects of steroids on ovarian development of spotty wrasse, or fish in general, in vitro. Although culture was associated with a degree of PVO degeneration, it was shown that administration of at least 10 ng/mL $E_2$ can partially mitigate tissue degeneration during culture. However, neither 11KT nor CORT treatment successfully accelerated ovarian atresia or induced spermatogenic proliferation. Future in vitro manipulations should explore longer cultures and/or co-treatment of multiple steroids to analyse potential antagonistic or synergistic effects. This culture system will facilitate future investigations of ovarian and testicular trans-differentiation in sex-changing fish. For example, an in vitro system would be especially amenable to the use of small interfering RNAs to knock down expression of specific candidate genes or application of methylation inhibitors to further investigate the genetic and epigenetic cascade underlying sex change. In vitro approaches also offer valuable ethical benefits over experiments with live fish, enabling the application of the 3Rs guiding principles: reduce, replace and refine (*Russell & Burch, 1959*).

### Abbreviations

| | |
|---|---|
| **11KT** | 11-ketotestosterone |
| **C** | control (no steroid) |
| **D0** | day 0 reference tissues |
| **CORT** | cortisol |
| **E2** | 17$\beta$-estradiol |
| **HDF5** | hierarchical data format 5 |
| **IP** | initial phase |
| **PVO** | previtellogenic oocyte |
| **TP** | terminal phase |

## ACKNOWLEDGEMENTS

We thank Sam Karelitz for assistance with bioinformatic analysis and support collecting samples. We are grateful to Helen Taylor for her help with sampling, the Portobello Marine Laboratory staff for kindly allowing us to use their facilities, and members of the Lokman

and Gemmell Labs for support. Sheri Johnson, Clare Holleley and Takashi Todo provided critical feedback on an earlier draft.

### Funding

Alexander Goikoetxea was supported by a scholarship from the Department of Anatomy, University of Otago. Erica V. Todd held a Health Sciences Career Postdoctoral Fellowship and Rutherford Postdoctoral Fellowship that supported this work. This work was further supported by a University of Otago Research Grant (ORG 0117-0318) to Neil J. Gemmell and Erica V. Todd, and a Marsden Grant (UOO1308) to Neil J. Gemmell and a Research Enhancement Grant (PBRF-ML63) to P. Mark Lokman. The funders had no role in study design, data collection and analysis, decision to publish, or preparation of the manuscript.

### Grant Disclosures

The following grant information was disclosed by the authors:
The Department of Anatomy, University of Otago.
Health Sciences Career Postdoctoral Fellowship.
Rutherford Postdoctoral Fellowship.
University of Otago Research: ORG 0117-0318.
Marsden: UOO1308.
Research Enhancement: PBRF-ML63.

### Competing Interests

The authors declare there are no competing interests.

### Author Contributions

- Alexander Goikoetxea conceived and designed the experiments, performed the experiments, analyzed the data, prepared figures and/or tables, authored or reviewed drafts of the paper, and approved the final draft.
- Erin L. Damsteegt, Erica V. Todd, Neil J. Gemmell and P. Mark Lokman conceived and designed the experiments, authored or reviewed drafts of the paper, and approved the final draft.
- Andrew McNaughton analyzed the data, authored or reviewed drafts of the paper, and approved the final draft.

### Animal Ethics

The following information was supplied relating to ethical approvals (i.e., approving body and any reference numbers):

Fish were captured and manipulated with approval from the University of Otago Animal Ethics Committee (AUP-18-247), and in accordance with New Zealand National Animal Ethics Advisory Committee guidelines.

### Data Availability

The raw measurements are available in the Supplementary Files.

## Supplemental Information

Supplemental information for this article can be found online at http://dx.doi.org/10.7717/peerj.10323#supplemental-information.

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
