# Peer review of "An in vitro ovarian explant culture system to examine sex change in a hermaphroditic fish"

_PeerJ, doi:10.7717/peerj.10323_

## Round 0.1 · original submission · Major Revisions

The paper focuses on a very interesting topic in order to try to establish an in vitro culture system for ovarian tissue of protogynous wrasse. The culture system is for 21 days (could be considered non-stable among the different specimens). Furthermore, only histological data have been used in this study and other techniques should be considered in order to better demonstrate the results.

Besides this, some references are needed in order to reinforce many sentences of the manuscript. For all these reasons, this paper seems to be a preliminary version that needs to be deeply improved.

Reviewer 1 ·

Basic reporting

no comment

Experimental design

no comment

Validity of the findings

no comment

Additional comments

In this report, authors try to establish an in vitro culture system for ovarian tissue of protogynous wrasse. This culture system (for 21 days) is used to evaluate the effects of different steroids on female-to-male sex change. Unfortunately, only histological data used in this study. It is very difficult to observe the differences on cellular and molecular levels.

1. For cell proliferating activity, IHC staining with a proliferating marker PCNA or proliferating cells labeling with the BrdU could help authors to evaluate the effect of 11KT and cortisol.
2. For male differentiation, several sex-dimorphic expression genes (Dmrt1, Amh, Gsdf) could help authors to evaluate the effect of 11KT and cortisol.
3. How authors to define the difference between the oogonia and spermatogonia in bisexual gonad? Authors have to add the figure to descript the characteristics of oogonia and spermatogonia in the early stage of sex change.
4. In Figure 2, the letters used to indicate significant difference is very difficult to understand.

Reviewer 2 ·

Basic reporting

1) The reviewer feels that the cited references are slightly less. Adding more references might be informative for the Journal readers. For example at line 58 and 60.

2) line 74: The reference 11 is focused on in vitro culture for kidney development. And the reference 10 does not mention in vitro culture in detail. The reviewer feels that more suitable reference should be cited if possible.

3) line 83 and line 218: The reference 12 seems to be doctoral thesis. The reviewer is not sure that this reference is appropriate.

4) line 104: Your introduction regarding the roles of cortisol need more detail. The roles on sex differentiation in gonochoristic species and the roles on sex changing species should be introduced, briefly. This might be informative for the Journal readers.

5) line 212 : Regarding the effect of 11KT, information on the related wrasse (J Exp Zool A Ecol Genet Physiol. 2012 : 317:552-60. ) might be helpful for discussion.

Experimental design

1) line 137: Your organ culture technique in materials and methods need more detailed explanation. Information on the amount of medium in each well and the frequency of medium exchange would be important information. Those should be mentioned.

2) For image analysis using Ilastik , information on sizes of previtellogenic oocytes would be important.
Usually, ovarian tissue has various sizes of previtellogenic oocytes. Therefore, the reviewer is wondering whether previtellogenic oocytes at smaller size can be distinguished by Ilastik software?

Validity of the findings

no comment

Additional comments

Some histological photos at high resolution might be helpful to see the size of cells. Those photos are also necessary to understand whether non atretic cells are normal or not. Photos of D0 tissue might be needed, too.
The reviewer would like to know if the sizes of remaining oocytes differ between each steroid treatment.

---

## Round 0.2 · Major Revisions

Thank you for improving your manuscript. However, there is still a major concern related to the results of the Ilastik analysis for the small sizes of oocytes and the stages of atretic cells have not been clear. Therefore, some additional photos to show these histological and/or cellular information are needed.

Reviewer 1 ·

Basic reporting

no comment

Experimental design

no comment

Validity of the findings

no comment

Additional comments

In this report, authors try to establish a ovarian culture system of wrasse and then use for the future study. However, authors only used image software to validate the ovarian development. It is hard to understand the relation between the oocyte size and cellular situation. Thus, reviewer believes that cell proliferating assay (IHC staining with a proliferating marker PCNA or proliferating cells labeling with the BrdU) and apoptosis assay (TUNEL assay) would help to improve the quality of this study. In general, the commercial anti-PCNA antibody can widely use for many fish species.

Reviewer 2 ·

Basic reporting

no comment

Experimental design

The reviewer understands the Ilastik software can be used to distinguish the cells in gonads. This may be the first trial. Therefore, the reviewer would like you to recommend to mention about the data settings for Ilastik more carefully. Does the data vary when the different sizes of previtellogenic oocytes and/or the different stages of atretic cells are labeled to train Ilastik ? It would be more helpful to understand the availability of Ilastik if you show these kinds of data.

Validity of the findings

no comment

Additional comments

The reviewer would like to express his respect for your efforts to revise the manuscript.
Overall, the manuscript seems to be much improved.
On the other hand, information on the cellular and histological stages of oocytes may be still unclear, although the resolution of pictures has been improved.
As the reviewer might say at the previous comments, the results of the Ilastik analysis for the small sizes of oocytes and the stages of atretic cells has not been clear. Therefore, it may be better to use some additional photos to show these histological and/or celllular information.
Moreover, in this manuscript, it might be important to understand the differences of the cellular stages between in vivo and in vitro. The reader would like to know how different between the initial tissue and the cultured tissue. In addition, the analyses of histological and/or cellular differences of gonadal change between the present in vitro data and the previous in vivo data during gonadal sex change might be informative.

---

## Round 0.3 · accepted · Accept

I am pleased to confirm that your paper has been accepted for publication in PeerJ.

Thank you for submitting your work to this journal.